# A scoping review protocol on brain PaCO$_2$ levels at altitude

**Hanna Tang** [1,2]* , **Laurel Charlesworth** [1,2,3] , **Manoj Lalu** [3,4] , **Brian Dewar** [3] , **Risa Shorr** [5] , **Dariush Dowlatshahi** [1,2,3]

1 Division of Neurology, Department of Medicine, The Ottawa Hospital, Ottawa, Ontario, Canada, 2 Faculty of Medicine, University of Ottawa, Ottawa, Ontario, Canada, 3 Ottawa Hospital Research Institute, Ottawa, Ontario, Canada, 4 Department of Anesthesiology and Pain Medicine, Department of Cellular and Molecular Medicine, University of Ottawa, Ottawa, Ontario, Canada, 5 Learning Services, The Ottawa Hospital, Ottawa, Ontario, Canada

☯ These authors contributed equally to this work.
* hatang@toh.ca

**Data Availability Statement:** No datasets were generated or analysed during the current study. All relevant data from this study will be made available upon study completion.

## Abstract

### Background

Aeromedical transfer of patients with ischemic stroke to access hyperacute stroke treatment is becoming increasingly common. Little is known about how rapid changes of altitude and atmospheric pressure can impact cerebral perfusion and ischemic burden. In patients with ischemic stroke, there is a theoretical possibility that this physiologic response of hypoxia-driven hyperventilation at higher altitude can lead to a relative drop in PaCO$_2$. This would ultimately result in cerebral vasoconstriction, and therefore worsening of the ischemic burden in patients with ischemic stroke. Currently, there are no specific recommendations in stroke guidelines for optimizing altitude of aeromedical transportation to minimize the rate of ischemic burden. In this scoping review, we aim to map the available literature that addressed whether PaCO$_2$ changes with altitude. This would be the steppingstone for more in-depth analyses into the cerebral autoregulatory consequences of high altitude on cerebral ischemia to form future guidelines.

### Methods and analysis

We will follow scoping review methods recommended by the Joanna Briggs Institute. Electronic databases MEDLINE, Embase, Web of Science, and Cochrane Central Register of Systematic will be systematically searched to identify articles that report on the acute response of PaCO$_2$ on acute change in altitude. Two independent investigators will perform duplicate title and abstract screening and full-text review, followed by duplicate data extraction. We will present quantitative data using descriptive statistics. Key textual information will be synthesized in a tabular format Simple statistics on the frequency of papers, data will be presented via histogram.

### Ethics and dissemination

This scoping review does not require ethical approval. The results of our scoping review will be published in academic medical journals and presented at medical conferences. The findings will highlight the current availability of literature on PaCO$_2$ changes with altitude.

**Funding:** The author(s) received no specific funding for this work.

**Competing interests:** The authors have declared that no competing interests exist.

## Registration

This scoping review protocol has been registered in Open Science Framework (DOI 10.17605/OSF.IO/UVK83).

## Introduction

Aeromedical transfer of patients with ischemic stroke allows access to hyperacute therapies that are not otherwise geographically available [1, 2]. Since 2018, when the Thrombectomy 6 to 24 Hours after Stroke with a Mismatch between Deficit and Infarct Trial demonstrated a 90-day mortality and recovery benefit with endovascular therapy (EVT) in patients who present with acute stroke within 24 hours of onset [3], the treatment window for ischemic stroke was extended to 24 hours from last seen well. This led to an expansion of catchment areas of EVT-capable comprehensive stroke centres. To this end, aeromedical transport is increasingly used worldwide to reduce the overall time to treatment for patients with strokes suspected to be secondary to a large vessel occlusion [4]. However, little is known about how rapid changes of altitude and atmospheric pressure can impact cerebral perfusion and ischemic burden.

At sea level, ventilation is driven by predominantly by arterial pressure of carbon dioxide (PaCO$_2$) with a secondary ventilatory response to hypoxia with lower arterial pressures of oxygen (PaO$_2$). The partial pressure of oxygen (pO$_2$) decreases at higher altitudes when atmospheric pressure decreases, leading to relative hypoxia; this hypobaric hypoxic environment causes hypoxia-driven hyperventilation, in an attempt to normalize PaO$_2$ [5, 6]. In patients with ischemic stroke, there is a theoretical possibility that this physiologic response of hypoxia-driven hyperventilation at higher altitude can lead to a relative drop in PaCO$_2$. There is a nearly direct linear relationship between PaCO$_2$ and cerebral autoregulation, with hypocarbia causing vasoconstriction. In the hypobaric hypoxic environment, hypoxia-drive hyperventilation lowers PaCO$_2$ which could lead to decreased cerebral blood flow and therefore worsening of the ischemic burden in patients with ischemic stroke. Although the use of supplemental oxygen could mitigate some of the effects of the hypobaric hypoxic environment during aeromedical transport, there are no current protocols for how this should be implemented, and it is currently unknown what blood oxygen saturation (spO$_2$) should be targeted. Furthermore, in patients with concurrent diseases impacting blood oxygenation, supplemental oxygen alone may not be sufficient to optimize PaO$_2$ and could require modifications in the cabin pressure and cruising altitude which could delay air transfer.

Currently, there are no specific recommendations in stroke guidelines for optimizing altitude of aeromedical transportation to minimize the rate of ischemic burden [7–11]. However, before these guidelines can be established, there are some knowledge gaps that will need to be addressed. The physiologic relationship between PaO$_2$, PaCO$_2$, altitude and cerebral autoregulation during aeromedical transfer is not well known. In this scoping review, we aim to map the available literature that addresses whether PaCO$_2$ changes with altitude. This scoping review will provide essential foundational information for future investigations into the cerebral autoregulatory consequences of high altitude on cerebral ischemia.

## Methods

This study will be conducted based on the guidelines of the Joanna Briggs Institute (JBI) Methodology for Scoping Reviews [12]. We registered our protocol on the Open Science

Framework (DOI 10.17605/OSF.IO/UVK83). The study team includes individuals with expertise in acute stroke care, occupational and aviation neurology, methodology and library sciences.

## Review question

Does blood PaCO$_2$ change acutely with exposure to high altitudes in humans, and if so, does this occur at cruising altitudes used in aeromedical transportation?

## Inclusion criteria

**Participants.** We will include all human studies since we hope to eventually apply these results to inform our decision making for the aeromedical transportation of patients with stroke. We will include studies looking at both healthy participants and those with medical co-morbidities.

**Concept.** The scoping review will explore whether in humans, PaCO$_2$ levels change with acute exposure to high altitudes. There is no standardized flight altitude for air transportation of stroke patients in Canada, however most medivac providers will pressurize the cabin between 2,000 to 4,000 feet, and occasionally up to 8,000 feet (standard airline cabin pressure) depending on the medical condition of the patient. This is dependent on many factors such as aircraft specifications, distance and geography, the patient's medical condition, and flying conditions, among others. Higher cabin pressure is generally preferred from a medical perspective since it creates a relatively less hypoxic hypobaric environment. However, it has poorer fuel economy and can lead to delays for refueling when transfer occurs over long distances. With this context, we will aim to include studies where the observed change in altitude is at least 1000 feet. We will include studies that comment on PaCO$_2$ measured both directly via arterial blood gas and indirectly.

**Context.** The included studies must assess PaCO$_2$ at a minimum of two altitudes to comment on how altitude may impact PaCO$_2$ levels.

**Types of evidence sources.** We will include all articles in peer-reviewed published journals. This scoping review will include all randomized controlled trials, non-randomized controlled trials, analytical observational studies including prospective and retrospective cohort studies, case-control studies and analytical cross-sectional studies, descriptive observational study designs including case series, individual case reports and descriptive cross-sectional studies for inclusion. Qualitative studies will be excluded.

**Exclusion criteria.** Abstracted, letters, protocols and other publications that do not have full-text articles associated with them will be excluded. Additionally, this study will exclude studies looking at those under 18 years of age. Studies that only report on PaO$_2$, or report on PaCO$_2$ but do not report on both altitude and PaCO$_2$ levels will be excluded, as will studies that report on the PaCO$_2$ at altitudes of under 1000 feet. Finally, studies which are not published in English will be excluded.

**Search strategy.** We will design a search strategy of Ovid MEDLINE, Embase, Web of Science, and Cochrane Database of Systematic Reviews with the assistance of our team's information specialist (RS). The search terms were decided upon in consultation with subject matter experts as well as examining a body of literature pre-identified to be relevant to the research question. To identify studies that were not captured within the search, we will hand-search reference lists of included articles and consult with experts in the field. A draft search strategy is included in S1 Appendix. Our search strategy will undergo Peer Review of Electronic Search Strategy to ensure comprehensiveness [13].

**Evidence selection.** Our final list of search results will be uploaded in Covidence systematic review platform (Covidence, Australia), a cloud-based platform for scoping reviews. Following deduplication, two reviewers will independently screen the title and abstracts against the inclusion/exclusion criteria above. We will begin with a pilot exercise of screening a random sample of 25 titles/abstracts and assessing the agreement between these two reviewers. Following this trial, the team will meet to discuss any disagreements and make necessary modifications to the screening criteria to increase inter-reviewer agreement. Screening will only commence upon reaching 75% or greater agreement between reviewers. After title and abstract screening has been completed, full-text copies of the articles will be retrieved and screened. For both title and abstract and full-text screening, disagreements between reviewers will be adjudicated by a third reviewer.

**Data extraction.** Data extraction will be performed by two independent team members utilizing Covidence software. Prior to extraction beginning, a comprehensive data extraction form will be developed and then entered into the software to enable electronic data capture. This data extraction framework will be piloted by two reviewers on a corpus of 10 articles to ensure agreement between extractors. Following the pilot, the team will discuss discrepancies and clarify terminology to ensure that extraction is applied consistently across all studies. After the pilot process, each reviewer will extract data using the piloted extraction form. Following each reviewer's completion of extraction, a third reviewer will compare the two extraction forms and adjudicate any discrepancies. Upon satisfactory resolution of discrepancies, a finalized dataset will have been produced to be used for synthesis and analysis. The list of items to be extracted is listed in Table 1. This list will be expanded in an iterative fashion as appropriate during the execution of the review.

**Data analysis and presentation.** As this is not a meta-analysis, no pooling of data and accompanying statistical treatment will be performed. Simple statistics on the frequency of papers, data and accompanying information will be presented via histogram. Correlations between altitude and blood $PaCO_2$ levels will be presented graphically, as will any findings surrounding study methodology and results.

**Deviations from protocol.** We will adopt a traditional flexible and iterative approach to this scoping review. As such, all changes to our protocol will be carefully detailed in updates to our registration in Open Science Framework.

**Strengths and limitations.** The potential strengths of this review include bringing significant subject matter expertise to this area. Further, this review presents an opportunity to engage with a potentially new area of study within the context of multiple advances in acute

**Table 1. Variables for data extraction*.**

| Items to be extracted: | |
|---|---|
| 1. | Author(s) |
| 2. | Year of publication |
| 3. | Origin/country of origin (where the source was published or conducted) |
| 4. | Aims/purpose |
| 5. | Population and sample size |
| 6. | Methodology/methods |
| 7. | Altitude |
| 8. | Reason for altitude (flight, mountain, etc.) |
| 9. | $PaCO_2$ blood concentration |

*If there is a pressurized cabin system, we will add a "pressurized altitude" variable.

stroke care. It is unclear if altitude of aeromedical transportation would affect the rate of ischemic burden. If there is evidence that suggests that $PaCO_2$ is impacted at cruising altitudes of aeromedical transport, then it opens the door to further studies in this area.

A potential weakness of this project may be a small body of research to access, and research within that corpus may be small, biased, or otherwise non-generalizable. Our preliminary search result yielded articles with various formats of high altitude exposure such as air transport, mountain climbing, and residing at high altitudes. One way we aim to address this weakness and make it more applicable to our clinical context is to ensure that $PaCO_2$ was measured in at least two altitudes. Depending on the final list of included articles, it is not known if there will be sufficient evidence to comment on how brief exposures to high altitude impacts PaCO2. However, this could also in turn open the door to future studies to fill the potential knowledge gap.

**Dissemination.** The findings of this study will be reported using the Preferred Reporting Items for Systematic Reviews and Meta-Analyses extension statement for reporting of Scoping Reviews (PRISMA-SCR) [14]. We will aim to publish in academic medical journals and present at relevant scientific conferences, with a goal to inform clinical practice guidelines and aeromedical transport protocols.

## Supporting information

**S1 Checklist. PRISMA-P (Preferred Reporting Items for Systematic review and Meta-Analysis Protocols) recommended items to address in a systematic review protocol.** (DOCX)

**S1 Appendix. Completed Peer Review of Electronic Search Strategies (PRESS) worksheet.** (DOCX)

## Author Contributions

**Methodology:** Manoj Lalu, Brian Dewar, Risa Shorr.

**Supervision:** Dariush Dowlatshahi.

**Writing – original draft:** Hanna Tang.

**Writing – review & editing:** Laurel Charlesworth, Manoj Lalu, Dariush Dowlatshahi.

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
