## [Decision Letter · Decision Letter 0]

4 Nov 2024

PONE-D-24-34309A scoping review protocol on brain PCO2 levels at altitudePLOS ONE

Dear Dr. Tang,

Thank you for submitting your manuscript to PLOS ONE. After careful consideration, we feel that it has merit but does not fully meet PLOS ONE’s publication criteria as it currently stands. Therefore, we invite you to submit a revised version of the manuscript that addresses the points raised during the review process.

We look forward to receiving your revised manuscript.

Kind regards,

Masaki Mogi

Academic Editor

PLOS ONE

Journal Requirements:

1. When submitting your revision, we need you to address these additional requirements. Please ensure that your manuscript meets PLOS ONE's style requirements, including those for file naming. The PLOS ONE style templates can be found at https://journals.plos.org/plosone/s/file?id=wjVg/PLOSOne_formatting_sample_main_body.pdf and https://journals.plos.org/plosone/s/file?id=ba62/PLOSOne_formatting_sample_title_authors_affiliations.pdf 2. Please provide a complete Data Availability Statement in the submission form, ensuring you include all necessary access information or a reason for why you are unable to make your data freely accessible. If your research concerns only data provided within your submission, please write "All data are in the manuscript and/or supporting information files" as your Data Availability Statement. 3. Please include captions for your Supporting Information files at the end of your manuscript, and update any in-text citations to match accordingly. Please see our Supporting Information guidelines for more information: http://journals.plos.org/plosone/s/supporting-information.

Additional Editor Comments:

The manuscript has been evaluated by two reviewers. See the suggestions carefully and respond them appropriately.

Reviewers' comments:

Reviewer's Responses to Questions

**Comments to the Author**

1. Does the manuscript provide a valid rationale for the proposed study, with clearly identified and justified research questions?

Reviewer #1: Yes

Reviewer #2: Yes

2. Is the protocol technically sound and planned in a manner that will lead to a meaningful outcome and allow testing the stated hypotheses?

Reviewer #1: Yes

Reviewer #2: Yes

3. Is the methodology feasible and described in sufficient detail to allow the work to be replicable?

Reviewer #1: Yes

Reviewer #2: Yes

4. Have the authors described where all data underlying the findings will be made available when the study is complete?

Reviewer #1: Yes

Reviewer #2: Yes

5. Is the manuscript presented in an intelligible fashion and written in standard English?

Reviewer #1: Yes

Reviewer #2: Yes

6. Review Comments to the Author

You may also provide optional suggestions and comments to authors that they might find helpful in planning their study.

Reviewer #1: Tang and colleagues described a scoping review protocol that maps the current literature to determine if blood arterial pressure of CO2 (PaCO2) acutely changes at high altitudes during aeromedical transportation. The authors provide details of how the study will be conducted, including inclusion and exclusion criteria, search strategy, evidence selection, data extraction and analysis. Overall, this protocol could be novel to understanding the PaCO2 changes in patients with ischemic stroke during aeromedical transportation.

Points of concern are outlined below:

1.Helicopter cabins are usually not pressurized. Since cabin pressure is not considered an inclusion criteria factor, the authors should include a short discussion on how cabin pressure may affect the PaCO2.

2.The authors mention in lines 173-175 that the current protocol may have a potential limitation in that the number of related studies may be small. Do the authors have an estimated number of literatures before and after the screening?

3.Please provide a discussion session that expands on the current “Strengths and Limitations” and briefly elaborates on factors like high altitude exposure format (helicopter, airliners, mountain climbing, chamber), exposure time, etc, that could complicate data interpretation.

4.Keep the PaCO2 acronym consistent.

Reviewer #2: Title: A scoping review protocol on brain PCO2 levels at altitude

Here is the review of the manuscript by Hanna Tang et al., which is a scoping review protocol on brain PCO₂ levels at altitude in critically ill patients. The authors address an interesting and clinically relevant question about the potential effects of rapid changes in altitude and atmospheric pressure on cerebral perfusion and ischemic burden during aeromedical transport of patients with ischemic stroke. This is a well-written and engaging scoping review protocol that clearly outlines the objectives, methods, and reporting approach for the review.

That said, I have a few minor suggestions for improvement:

Exclusion Criteria: The authors indicate that they will exclude studies involving participants under 18 years of age. However, it is unclear why this exclusion criterion is necessary for their review.

Measurement of PaCO₂: The authors specifically aim to include studies that report arterial partial pressure of CO₂ (PaCO₂) at two different altitudes. Since studies may measure PaCO₂ either directly (e.g., via arterial blood gas) or indirectly, the authors should clarify their criteria regarding acceptable methods for PaCO₂ measurement.

7. PLOS authors have the option to publish the peer review history of their article (what does this mean?). If published, this will include your full peer review and any attached files.

Reviewer #1: No

Reviewer #2: No

---

## [Author Response · Author response to Decision Letter 0]

23 Nov 2024

Response to Reviewers:

Reviewer 1:

Comment 1: Helicopter cabins are usually not pressurized. Since cabin pressure is not considered an inclusion criteria factor, the authors should include a short discussion on how cabin pressure may affect the PaCO2.

Response: This is an excellent point. Although rotary wing transfers are the most commonly used airframe for aeromedical transfers and are often unpressurized, fixed wing transport at higher altitudes is required for longer distances; fixed wing cabins for medical transfer are invariably pressurized, often to an equivalent altitude 2,000-8,000 feet. We have included the following in our concepts section under inclusion criteria: “Higher cabin pressure is favoured from a medical perspective for the patient since it creates a relatively less hypoxic hypobaric environment.” The trade-off is that a pressurized cabin will increase fuel usage and on some flight distances, result in longer flight times due to the need for refueling. We have not included cabin pressure as an inclusion criteria since we would like to first scope all available literature in the field to see if altitude affects PaCO2, and since cabins are normally pressurized to over 1000 feet. If there is a pressurized cabin system, we will note in our review the equivalent altitude to which the cabin was pressurized. 

Comment 2: The authors mention in lines 173-175 that the current protocol may have a potential limitation in that the number of related studies may be small. Do the authors have an estimated number of literatures before and after the screening?

Response: Based on the initial search, we have found 491 articles prior to screening. We are not yet able to estimate the number of studies after screening, as we are awaiting approval of the final protocol before screening to maintain methodological rigor. 

Comment 3: Please provide a discussion session that expands on the current “Strengths and Limitations” and briefly elaborates on factors like high altitude exposure format (helicopter, airliners, mountain climbing, chamber), exposure time, etc, that could complicate data interpretation.

Response: We have elaborated on the “Strengths and Limitations” section. The following was added to the protocol: “It is unclear if altitude of aeromedical transportation would affect the rate of ischemic burden. If there is evidence that suggests that PaCO2 is impacted at cruising altitudes of aeromedical transport, then it opens the door to further studies in this area… Our preliminary search result yielded articles with various formats of high altitude exposure such as air transport, mountain climbing, and residing at high altitudes. One way we aim to address this weakness and make it more applicable to our clinical context is to ensure that PaCO2 was measured in at least two altitudes. Depending on the final list of included articles, it is not known if there will be sufficient evidence to comment on how brief exposures to high altitude impacts PaCO2. However, this could also in turn open the door to future studies to fill the potential knowledge gap.”

Comment 4: Keep the PaCO2 acronym consistent.

Response: Thank you for bringing this to our attention. This has been corrected. 

Reviewer 2: 

Comment 1: Exclusion Criteria: The authors indicate that they will exclude studies involving participants under 18 years of age. However, it is unclear why this exclusion criterion is necessary for their review.

Response: The majority of the Canadian Stroke Guidelines are for patients aged 18 and older. Ultimately the goal of this study is to provide future direction to improve our adult stroke care, which is the population who we serve clinically. Additionally, it is unclear if neonates and/or children would have the same hypoxic drive to breathe at altitude as in adults. But we appreciate the comment and agree after this initial review, our team will consider a similar assessment in the pediatric population.

Comment 2: Measurement of PaCO₂: The authors specifically aim to include studies that report arterial partial pressure of CO₂ (PaCO₂) at two different altitudes. Since studies may measure PaCO₂ either directly (e.g., via arterial blood gas) or indirectly, the authors should clarify their criteria regarding acceptable methods for PaCO₂ measurement.

Response: Very good point – we have now included the following statement in our inclusion criteria: “We will include studies that comment on PaCO2 measured both directly via arterial blood gas and indirectly.”

---

## [Decision Letter · Decision Letter 1]

15 Dec 2024

A scoping review protocol on brain PaCO2 levels at altitude

PONE-D-24-34309R1

Dear Dr. Tang,

We’re pleased to inform you that your manuscript has been judged scientifically suitable for publication and will be formally accepted for publication once it meets all outstanding technical requirements.

Kind regards,

Masaki Mogi

Academic Editor

PLOS ONE

Additional Editor Comments (optional):

Reviewers' comments:

Reviewer's Responses to Questions

**Comments to the Author**

1. Does the manuscript provide a valid rationale for the proposed study, with clearly identified and justified research questions?

Reviewer #1: Yes

Reviewer #2: Yes

2. Is the protocol technically sound and planned in a manner that will lead to a meaningful outcome and allow testing the stated hypotheses?

Reviewer #1: Yes

Reviewer #2: Yes

3. Is the methodology feasible and described in sufficient detail to allow the work to be replicable?

Reviewer #1: Yes

Reviewer #2: Yes

4. Have the authors described where all data underlying the findings will be made available when the study is complete?

Reviewer #1: Yes

Reviewer #2: Yes

5. Is the manuscript presented in an intelligible fashion and written in standard English?

Reviewer #1: Yes

Reviewer #2: Yes

6. Review Comments to the Author

You may also provide optional suggestions and comments to authors that they might find helpful in planning their study.

Reviewer #1: Tang and colleagues described a scoping review protocol that maps the current literature to determine if blood arterial pressure of CO2 (PaCO2) acutely changes at high altitudes during aeromedical transportation. The authors provide details of how the study will be conducted, including inclusion and exclusion criteria, search strategy, evidence selection, data extraction and analysis. Overall, this protocol could be novel to understanding the PaCO2 changes in patients with ischemic stroke during aeromedical transportation. The authors have addressed all my concerns. I don't have any further comments.

Reviewer #2: Here is the review of the manuscript by Hanna Tang et al., which is a scoping review protocol on brain PCO₂ levels at altitude in critically ill patients. The authors address an interesting and clinically relevant question about the potential effects of rapid changes in altitude and atmospheric pressure on cerebral perfusion and ischemic burden during aeromedical transport of patients with ischemic stroke. This is a well-written and engaging scoping review protocol that clearly outlines the objectives, methods, and reporting approach for the review. The concerns I had now well addressed and I suggest accepting the manuscript.

7. PLOS authors have the option to publish the peer review history of their article (what does this mean?). If published, this will include your full peer review and any attached files.

Reviewer #1: No

Reviewer #2: No

---

## [Editor Report · Acceptance letter]

24 Dec 2024

PONE-D-24-34309R1 

PLOS ONE

Dear Dr. Tang, 

I'm pleased to inform you that your manuscript has been deemed suitable for publication in PLOS ONE. Congratulations! Your manuscript is now being handed over to our production team.

Kind regards, 

on behalf of

Dr. Masaki Mogi 

Academic Editor

PLOS ONE